# Anemia or other comorbidities? using machine learning to reveal deeper insights into the drivers of acute coronary syndromes in hospital admitted patients

**Faisal Alsayegh**[1]☯, **Moh A. Alkhamis**[2]☯*, **Fatima Ali**[1], **Sreeja Attur**[1], **Nicholas M. Fountain-Jones**[3,4], **Mohammad Zubaid**[1]

1 Faculty of Medicine, Department of Medicine, Health Sciences Center, Kuwait University, Kuwait City, Kuwait, 2 Faculty of Public Health, Department of Epidemiology and Biostatistics, Health Sciences Center, Kuwait University, Kuwait City, Kuwait, 3 School of Natural Sciences, University of Tasmania, Hobart, Australia, 4 Department of Veterinary Population Medicine, College of Veterinary Medicine, University of Minnesota, St. Paul, Minnesota, United States of America

☯ These authors contributed equally to this work.
* m.alkhamis@ku.edu.kw

**Data Availability Statement:** All relevant data are uploaded to https://figshare.com/articles/dataset/Anemia_or_other_comorbidities_Using_machine_

## Abstract

Acute coronary syndromes (ACS) are a leading cause of deaths worldwide, yet the diagnosis and treatment of this group of diseases represent a significant challenge for clinicians. The epidemiology of ACS is extremely complex and the relationship between ACS and patient risk factors is typically non-linear and highly variable across patient lifespan. Here, we aim to uncover deeper insights into the factors that shape ACS outcomes in hospitals across four Arabian Gulf countries. Further, because anemia is one of the most observed comorbidities, we explored its role in the prognosis of most prevalent ACS in-hospital outcomes (mortality, heart failure, and bleeding) in the region. We used a robust multi-algorithm interpretable machine learning (ML) pipeline, and 20 relevant risk factors to fit predictive models to 4,044 patients presenting with ACS between 2012 and 2013. We found that in-hospital heart failure followed by anemia was the most important predictor of mortality. However, anemia was the first most important predictor for both in-hospital heart failure, and bleeding. For all in-hospital outcome, anemia had remarkably non-linear relationships with both ACS outcomes and patients' baseline characteristics. With minimal statistical assumptions, our ML models had reasonable predictive performance (AUCs > 0.75) and substantially outperformed commonly used statistical and risk stratification methods. Moreover, our pipeline was able to elucidate ACS risk of individual patients based on their unique risk factors. Fully interpretable ML approaches are rarely used in clinical settings, particularly in the Middle East, but have the potential to improve clinicians' prognostic efforts and guide policymakers in reducing the health and economic burdens of ACS worldwide.

learning_to_reveal_deeper_insights_into_the_
drivers_of_in-hospital_Acute_coronary_
syndromes/16715557.

**Funding:** Gulf COAST is an investigator-initiated study, financially supported by AstraZeneca and sponsored and overseen by Kuwait University (Project Code XX02/11). Neither Kuwait University nor AstraZeneca had any role in study design, data collection, data analysis or writing of the manuscript.

**Competing interests:** Further, all authors declare that they have no competing interests related to AstraZeneca or other commercial funders, since this study was conducted as a secondary analysis from earlier studies cited in the manuscript. All authors confirm that this does not alter their adherence to PLOS ONE policies on sharing data and materials as detailed on the journal guidelines.

## Introduction

Cardiovascular diseases are responsible for one-third of deaths worldwide, with projected mortalities of up to 7.8 million in 2025 [1]. Besides the costs of treatment and intervention programs, premature deaths due to cardiovascular disease cause substantial global economic losses due to lost productivity [2]. Effective primary prevention is often difficult due to the complexity of cardiovascular disease epidemiology and the dynamic nature of risk profiles that are rapidly changing in response to increasing urbanization and globalization and shifts in demography [3,4]. This is particularly true for Acute coronary syndromes (ACS) that is an important category of cardiovascular disease that includes unstable angina and myocardial infarction. The complexity of the epidemiology of ACS poses a significant challenge to the prognostic capacities of primary and secondary care clinicians leading to a higher frequency of negative in-hospital outcomes. Large population-level studies provide critical insights on ACS risk factors, however, non-linear relationships and complex interactions between ACS risk factors make inference and prediction difficult. Machine Learning (ML) algorithms can capture these complex relationships to build powerful predictive models that have provided important insights into the clinical epidemiology of cardiovascular diseases generally [5,6]. However, ML models are often considered 'black box' and can be difficult to interpret. Interrogating these 'black box' models with advances in interpretable machine learning can help gain mechanistic insights into predictions in a variety of systems (Molnar, 2018, Fountain-Jones et al). Interpretable machine learning methods, however, are rarely used to help predict and interpret ACS risk.

ACS diagnosis and treatment are significant challenges for clinicians due to the significant overlap in symptoms between ACS patients and non-patients [7]. Despite the availability of many ACS diagnostic tools (*e.g.*, coronary angiography, cardiac markers, and electrocardiographic), nearly two to five percent of ACS true cases were wrongly discharged from the emergency room due to the false indications of non-cardiac disease [8,9]. This diagnostic error is a leading cause of ACS mortalities worldwide, causing severe public health and economic implications [9]. However, the risk stratification approach [10], significantly helped clinicians improve their diagnostic and prognostic efforts of ACS events over the past few decades. Risk stratification is defined as formal prediction procedure of ACS events according to the individual patients' risk at the time of presentation [11], and include many tools such as Thrombolysis in Myocardial Infraction (TIMI) [12], the Evalution of Methods of Management of Acute Coronary Syndrome (EMMACE) [13,14], and the Global Registry of Acute Coronary Events (GRACE) [15–17]. These tools rely on calculations from data on patient's presenting symptoms, historical information available at the time of presentation, and laboratory result studies [12].

Anemia is one of the most observed comorbidities, with an estimated worldwide prevalence of 10%-43% in patients with ACS. Anemia can aggravate ACS outcomes due to the twin effect of the decreased overall oxygen content of the blood leading to ischemic myocardial tissue and subsequently increased cardiac output to sustain a sufficient oxygen supply [18]. Therefore, anemia is an independent predictor of adverse cardiovascular events in patients with ACS [18]. There is some evidence that anemia is strongly associated with severe hemorrhagic complications and short and long-term mortality in ACS [19]. Further, older ACS patients with chronic anemia typically suffer from other co-morbidities such as chronic heart failure and kidney disease when compared to their counterparts with normal hemoglobin levels [20]. Anemia can also compromise or delay interventional procedures such as coronary angiography and percutaneous coronary intervention (PCI), leading to potential cardiac complications [21,22]. While research on the relationship between anemia and ACS outcomes is limited in

the Middle East, the Gulf Registry of Acute Coronary Events–II (GULF RACE II) study found that nearly 28% of ACS patients were anemic at the time of admission [23]. Additionally, most of their anemic patients suffered from multiple in-hospital ACS related complications [23].

The inherent limitations of population-based studies from registries or clinical trials may partially pose an obstacle to improving ACS diagnostic and prognostic performance. For example, the generalizability of inferences from such studies may not realistically reflect all the patients with ACS or represents populations with special risk factors [24]. While risk stratification tools such as major adverse cardiovascular event (MACE) or Thrombolysis in Myocardial infarction (TIMI) can tailor personalized interventions based on individual patient-level predictions, they mainly depend on regression scoring systems that primarily assume linear relationships between the outcome and its predictors on a population level [11,12]. Additionally, traditional statistical linear models are susceptible to overfitting and tend to underperform with large datasets collected by registries, partially due to the high correlations between variables [25]. In contrast, ML algorithms require minimal statistical assumptions, can explore large data sets, and accommodate thousands of variables of different varieties (*e.g.*, genomic data, medical images). ML algorithms can also efficiently and robustly quantify complex interactions between variables providing the ability to infer novel insights into the clinical epidemiology of ACS. Further, ML algorithms can outperform traditional statistical methods in individual-level predictions, rendering them the most suitable tools for improving clinical performance [5,6]. Yet, ML algorithms have not been widely adopted in clinical practice, particularly in countries in the Middle East where ACS is common.

Here, we apply a newly developed multi-algorithm ML ensemble pipeline to the Gulf locals with ACS events (Gulf COAST) registry to identify which factors shaped the risk of different in-hospital ACS events among four Gulf countries. More specifically, we used patient characteristics data to build interpretable predictive risk models for three in-hospital ACS events, including heart failure, bleeding, and mortalities, to identify and compare their unique requirements for onset in a clinical setting. Further, we explore the role of patients' initial hemoglobin values upon admission (*i.e.*, admission anemia), and their interactions with other relevant factors for each selected ACS event. Moreover, we extend and evaluate our models in the context of in-hospital individual level prognosis to address the utility and limitations of interpretable ML models compared to traditional risk stratification methods.

## Methods

### Data source

We retrieved our data from the Gulf locals with acute coronary syndromes events (Gulf-COAST registry), which comprises 4,044 records of patients admitted with a diagnosis of ACS to 29 hospitals between January 2012 and January 2013 in Bahrain, Kuwait, Oman, and the United Arab Emirates. A detailed description of the design and implementation of the registry is available elsewhere [26]. We used the world health organization (WHO) definition of anemia in adults (Males <13 g/dl, Females <12 g/dl) in our study [27]. We selected variables (hereafter 'features') that were found to be significantly associated with ACS events including variables capturing patient demographics, past medical history, medical status upon admission and in-hospital ACS outcomes (Table 1) [28–30]. We used infarction/reinfarction, percutaneous coronary intervention (PCI), heart failure, stroke, bleeding, and mortality as independent outcomes for our predictive models (6 ML models in total). Also, we reused these in-hospital outcomes as independent features for predicting the risk of each corresponding ACS event (e.g., using in-hospital bleeding as an independent predictor of mortality).

**Table 1. Baseline characteristics of Gulf-COAST patients.**

| Study population | n = 4,044 (%) |
|---|---|
| *Demographics and hemoglobin on admission* | |
| Country | |
| UAE | 691 (17.14) |
| Kuwait | 1,230 (30.51) |
| Oman | 1,481 (36.74) |
| Bahrain | 629 (15.60) |
| Sex (Female) | 1,354 (33.59) |
| Age, mean ± SD (years) | 60.33 ± 12.69 |
| Anemia (at admission) | 1,713 (42.36) |
| Initial hemoglobin, mean ± SD (g/dl) | 13.26 (2.06) |
| Smoking | 1,593 (39.52) |
| Alcohol consumption | 126 (3.13) |
| Age, mean ± SD (years) | 60.33 ± 12.69 |
| *Past Medical History* | |
| Hypertension | 2,612 (64.80) |
| Dyslipidemia | 2,277 (56.49) |
| Diabetes mellitus | 2,179 (54.06) |
| Previous history of CVD | 2,354 (58.40) |
| Stroke or TIA[b] | 290 (7.19) |
| Chronic renal failure | 292 (7.24) |
| Cancer | 44(1.09) |
| *In-hospital outcomes* | |
| Infarction/reinfarction | 71 (1.76) |
| PCI[c] | 14 (0.35) |
| Heart Failure | 521 (12.92) |
| Stroke | 36 (0.89) |
| Bleeding | 119 (2.95) |
| Mortality | 167 (4.14) |
| Length of hospital stay, mean ± SD (days) | 5.90 (7.36) |
| *Prevalence of In-hospital ACS[d] per subtypes* | |
| LBBB MI | 3/30 (10.00) |
| STEMI | 436/997 (43.73%) |
| NSTEMI | 763/1,916 (39.82%) |
| Unstable Angina | 351/1,101 (31.88%) |

[a]Cardiovascular Disease;

[b]Transient Ischemic Attack;

[c]Percutaneous Coronary Intervention; ACS; Acute Coronary Syndrome.

## Data processing

We used the ML pipeline proposed by Fountain-Jones *et al.*, which constructs predictive models and compares four popular supervised algorithms, including random forest (RF), support vector machine (SVM), gradient boosting (GBM), and logistic regression (LR). ML algorithms construct classification models using different approaches (see Fountain-Jones *et al.*), and comparing performance across algorithms is important to optimize importance. We excluded features with the largest mean absolute correlation ($\rho > 0.9$) and applied the 'Boruta' R package to eliminate further features to just those relevant for prediction to boost the performance of our ML algorithms [31].

We controlled for class imbalance using a down-sampling procedure that randomly down samples the majority class to match its frequency to the minority class (*i.e.*, patients discharged with ACS events). We then randomly partitioned the dataset into a training (80%) and testing (20%) sets and used the K-fold cross-validation (K = 10) procedure to train the ML algorithms. All of our statistical analyses were conducted in the R software environment [32].

## Model training and evaluation

We trained our ML algorithms using the complete set of features for each ACS in-hospital event (Table 1). We ran the GBM, SVM, and LR algorithms using the 'Caret' R package while we used the 'random Forest' R package to run the RF algorithm [33–35]. We estimated the performance parameters of each model, including the area under the curve through a receiver operator characteristic (ROC), accuracy (Acc), specificity (Sp), and sensitivity (Se) using the 10-fold cross-validation approach. These parameters were calculated using the average confusion matrix across all folds of the cross-validation. Here, we used the 10-fold cross-validation procedure to avoid overfitting due to the use of the same data for training and validation, as well as to prevent artificial inflation of the accuracy. Default grid parameter settings were used in the training process of all algorithms. We then compared the estimated validation parameters of each model using the testing dataset to select the best performing algorithm in predicting the probability of an in-hospital ACS event.

## Model interpretation

We interrogated our best-performing models feature importance, partial dependence, feature interaction strength and the relationships between features and the ACS events on randomly selected individual patients. We used Breiman's permutation procedure to compute feature importance, which is implemented in the 'iml' R package [33,36]. This method quantifies the expected loss in predictive performance (*i.e.*, how the algorithm classifies the occurrence of patients ACS events) for a pair of observations compared to the full model when a specific feature has been switched [33,37]. Thus, the feature is deemed unimportant when the permutation procedure does not affect model performance. We used partial dependence (PD) plots and centred individual conditional expectation (ICE) to estimate the global and individual effects of each important feature on the response, and each observation, respectively [38]. Feature interaction strength was quantified using Friedman's H-statistic, which accounts for the portion of variance explained by the interaction through a partial dependency decomposition procedure [39]. Finally, following a game theory approach, we calculated Shapley values ($\phi$), from the final selected models. This unique approach quantifies individual-level predictions for randomly selected patients and the contribution of each feature to those predictions [40].

## Results

For both in-hospital mortality and bleeding models, the RF algorithm slightly outperformed other algorithms in terms of performance parameters (*i.e.*, AUC, Acc, Sp, Se) and correctly predicted 85% and 75% of death and bleeding events, respectively (AUCs = 0.85 & 0.75; Table 2). In contrast, the GBM algorithm slightly outperformed other algorithms and correctly predicted heart failure 77% of the time (AUC = 77.44; Table 2). While LR model consistently had poor performance (AUCs < 0.5; Table 2) and was leaning toward random prediction for all ACS outcomes. Also, for all in-hospital events models, LR performance parameters were notably lower than other ML algorithms (Table 2). Overall, the mortality model had the highest performance parameters when compared to both heart failure and bleeding models

**Table 2. Cross-validation summary results for GBM, RF and SVM models.**

| Model | AUC[a] ± SE[b] | Accuracy (%) ± SE | Specificity (%) ± SE | Sensitivity (%) ± SE |
|---|---|---|---|---|
| | | *Mortality* | | |
| RF[c] | 0.85 ± 0.02 | 79.18 ± 1.17 | 79.21 ± 1.19 | 74.15 ± 3.58 |
| GBM[d] | 0.81 ± 0.01 | 78.72 ± 2.12 | 78.73 ± 2.14 | 75.86 ± 2.95 |
| SVM[e] | 0.82 ± 0.00 | 76.29 ± 1.70 | 72.28 ± 1.75 | 79.54 ± 2.91 |
| LR[f] | 0.69 ± 0.00 | 60.19 ± 2.35 | 55.29 ± 1.99 | 69.71 ± 2.34 |
| | | *Heart Failure* | | |
| RF | 0.68 ± 0.00 | 62.24 ± 1.39 | 62.18 ± 1.42 | 65.92 ± 1.25 |
| GBM | 0.77 ± 0.01 | 68.76 ± 1.69 | 69.67 ± 1.74 | 70.66 ± 1.71 |
| SVM | 0.71 ± 0.02 | 62.55 ± 1.76 | 62.46 ± 1.82 | 67.50 ± 1.59 |
| LR | 0.58 ± 0.10 | 63.89 ± 2.66 | 61.79 ± 1.34 | 58.15 ± 1.88 |
| | | Bleeding | | |
| RF | 0.76 ± 0.01 | 70.57 ± 3.21 | 72.11 ± 3.23 | 57.81 ± 5.06 |
| GBM | 0.65 ± 0.02 | 64.58 ± 3.14 | 64.60 ± 3.16 | 58.64 ± 4.58 |
| SVM | 0.63 ± 0.00 | 77.06 ± 2.79 | 77.09 ± 1.75 | 66.87 ± 1.75 |
| LR | 0.54 ± 0.00 | 66.11 ± 4.33 | 55.78 ± 3.99 | 56.72 ± 3.83 |

[a]Area Under the Curve,

[b]Standard error,

[c]Random Forest,

[d]Gradient Boosting,

[e]Support Vector Machine,

[f]LR: Logistic Regression. Model highlighted in gray is the best performing model.

(Table 2). Our models were not able to accurately predict the need for PCI, stroke, and infarction models.

Our ML approach revealed that in-hospital heart failure followed by initial hemoglobin values at admission and age, were the most important features for predicting the risk of mortality (Fig 1A). Notably, hemoglobin was the most important predictor for both in-hospital heart failure and bleeding events (Fig 1B and 1C). However, in-hospital infraction and bleeding were the second and third important predictors for heart failure, respectively (Fig 1B). In contrast, age and heart failure were the second and third important predictors for bleeding, respectively (Fig 1C).

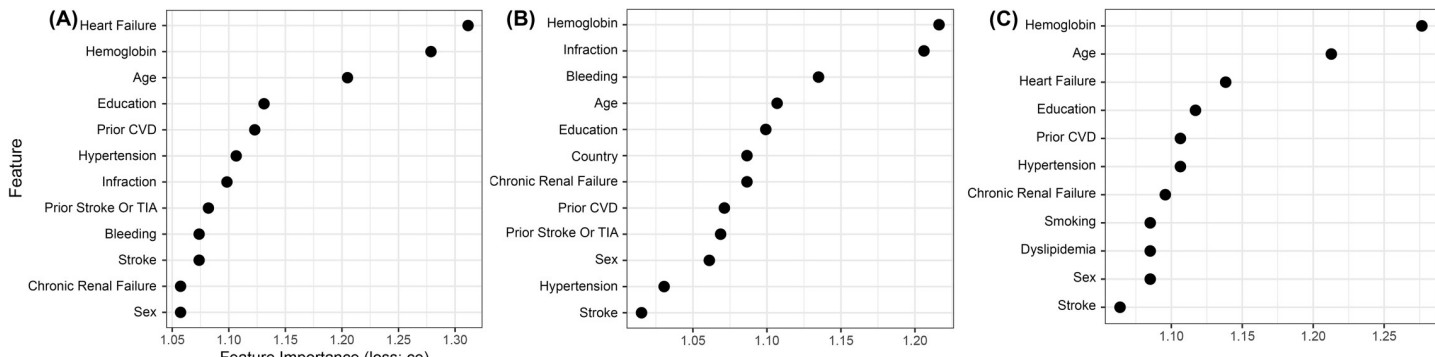

**Fig 1.** Important Features that Contribute to the Prediction of Three In-Hospital ACS Related Outcomes, Including (A) mortality, (B) heart failure, and (C) Bleeding. A classification error loss function ("ce") was used to calculate feature importance. Black dots indicate median "ce,". CVD: Cardiovascular disease. TIA: Transient ischemic attack.

PD plots consistently showed that the risk of all ACS outcomes increased when patients with initial hemoglobin values of equal or less than 10 g/dl are more likely to be discharged dead, had heart failure or developed bleeding (Fig 2B–2D). Our PD plots also show that patients with in-hospital heart failure (Fig 2A) and those aged over 75 years (Fig 2G) are more likely to be discharged dead. Further, patients having in-hospital infarction (Fig 2E) and bleeding (Fig 2H) are more likely to experience an in-hospital heart failure (Fig 2H). However, in

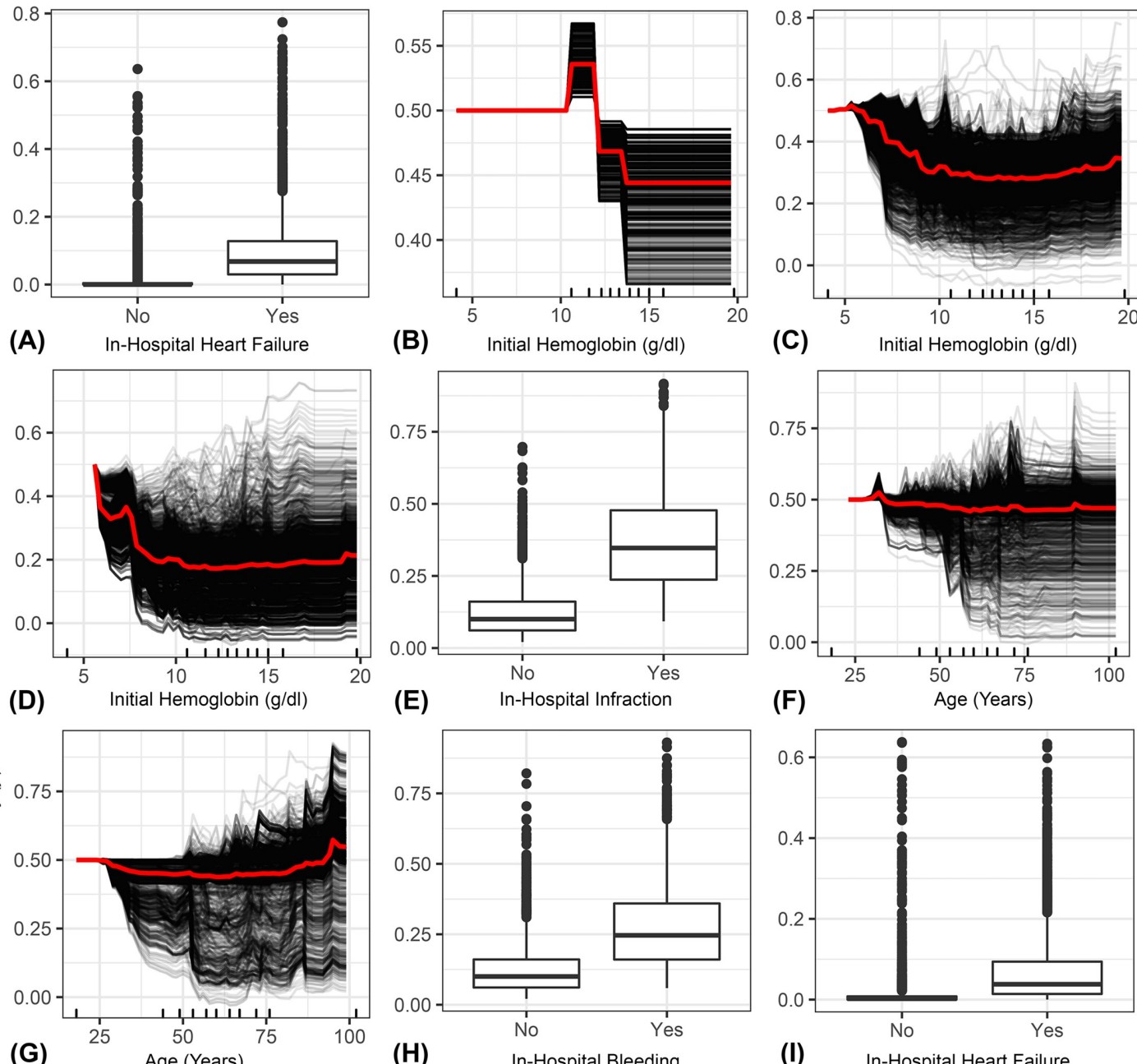

**Fig 2.** Centred Individual Conditional Expectation (ICE) plots for the top three important features that contribute to the prediction of three in-hospital ACS outcomes, including (A, D, G) mortality, (B, E, H) heart failure, and (C, F, I) bleeding. The plots show the relationship between the predicted risk of ACS outcomes and each corresponding feature. The black lines indicate the predicted risk in each patient, while the red line indicates the partial dependence calculated as the average risk across all patients.

the bleeding model the effect of age was inconclusive with no distinct trends (Fig 2F). Yet, patients having in-hospital heart failure are more likely to experience bleeding events (Fig 2I).

Initial hemoglobin on admission had the strongest interactions with the other features in shaping the risk of in-hospital deaths and heart failure events (Fig 3A and 3D). However, chronic renal failure had the strongest overall interaction strength, among other features for shaping the risk of bleeding (Fig 3G). Nevertheless, the interaction between admission hemoglobin and age (Fig 3B) was the strongest for predicting the risk of death from an ACS event, in which the majority of the patients aged above 75 years with hemoglobin values close to or less

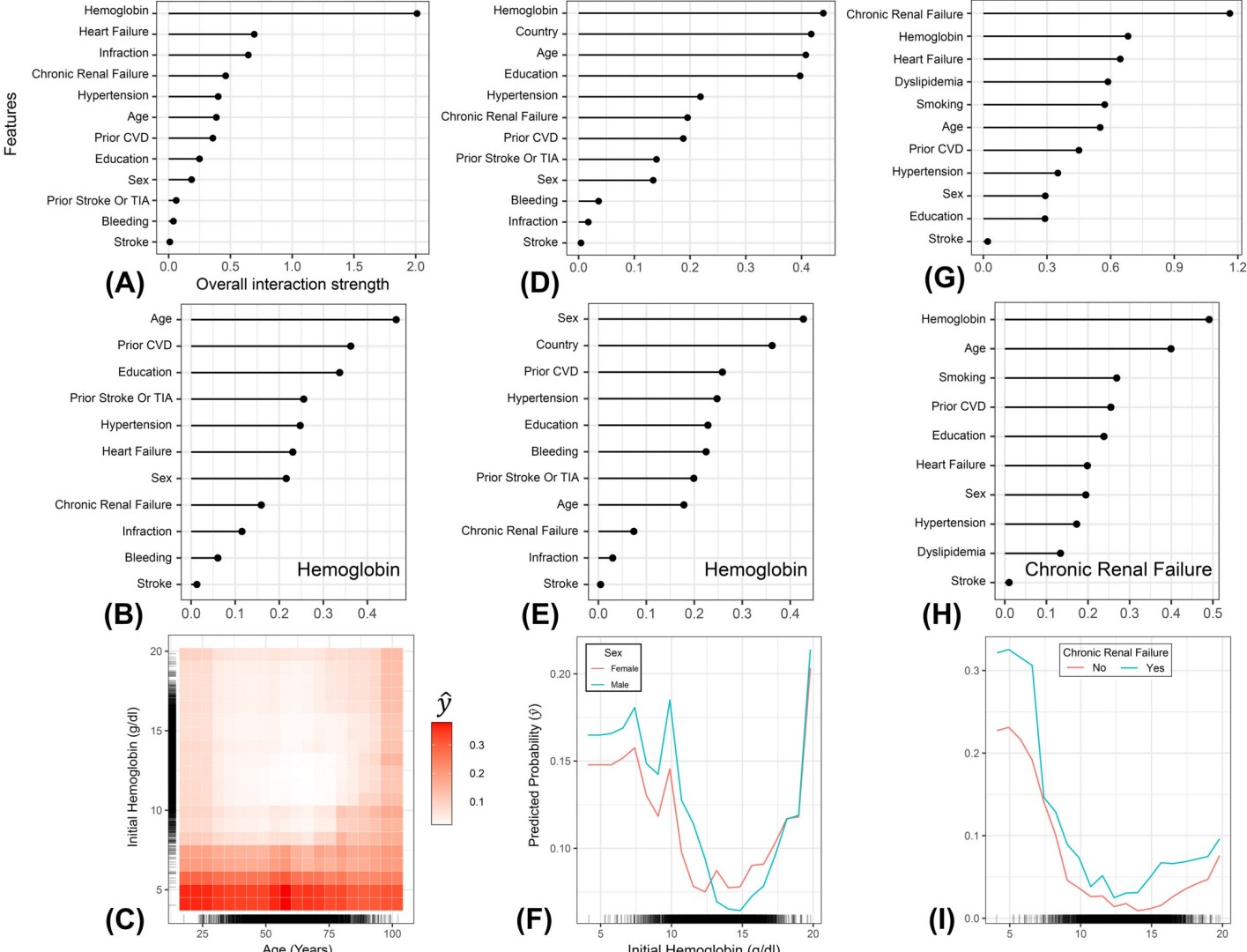

**Fig 3. Feature interaction plots calculated using Friedman's H-statistic.** (A-C) indicate the mortality model; (D-F) indicate the heart failure model; (G-I) indicate the bleeding model. The three plots on the top (A, D, G) showing the overall interaction strength of each feature with the other features. Plots (B & E) demonstrates the overall interaction strength of hemoglobin with the other features, while plot (H) demonstrate the overall interaction strength of chronic renal failure with the other features. Partial dependence plots at the bottom (C, F, I) represent the top interactions that shaped the risk of the three acute coronary syndromes outcomes. (C) interaction between age and initial hemoglobin for the mortality model. The heat matrix corresponds to the risk of death, in which lighter shades of red indicate lower risks of death, and darker shades of reds indicate higher risks of death. The bar on the right indicates (y hat) the relative risk of being dead with all other feature combinations marginalized. (F) interaction between sex and initial hemoglobin for the heart failure model. (I) interaction between chronic renal failure initial hemoglobin for the bleeding model. Red and green partial dependence curves represent different sexes (F) and the status of chronic renal failure (I), while the y-axis indicates the predicted risk of heart failure or bleeding. CVD: Cardiovascular disease. TIA: Transient ischemic attack.

than 10 g/dl are at high risk of death from an ACS event (Fig 3C). For the heart failure model, the interaction between admission hemoglobin and sex was the strongest among other interactions (Fig 3E). Further, notable declines in risk were inferred for both males and females at hemoglobin values greater than 10 g/dl, but with a sharp increase at hemoglobin values ≅ 20 10 g/dl (Fig 3F). However, the risk of heart failure in males at lower hemoglobin values was slightly higher than in females (Fig 3F). For the bleeding model, the interaction between hemoglobin and chronic renal failure was the strongest among other interactions (Fig 3H), in which anemic patients with chronic renal failure are more likely to experience bleeding events (Fig 3I).

The game-theoretic approach we used provided more insight into how the best performing model predicted ACS outcomes at an individual patient level (Fig 4). Patients with hemoglobin values of 10 g/dl or less were observed with ACS, and the models predicted that they are more likely to experience either death, heart failure, or bleeding (probabilities > 0.8; Fig 4A–4C). Conversely, patients with hemoglobin values higher than 10 g/dl were also observed, and the models predicted that they are less likely to experience either death, heart failure, or bleeding (probabilities = 0.0; Fig 4D–4F).

## Discussion

Using our integrated ML pipeline and the Gulf COAST data, we uncovered deeper insights into the factors that shape the outcome of ACS in the Gulf countries. We also revealed the

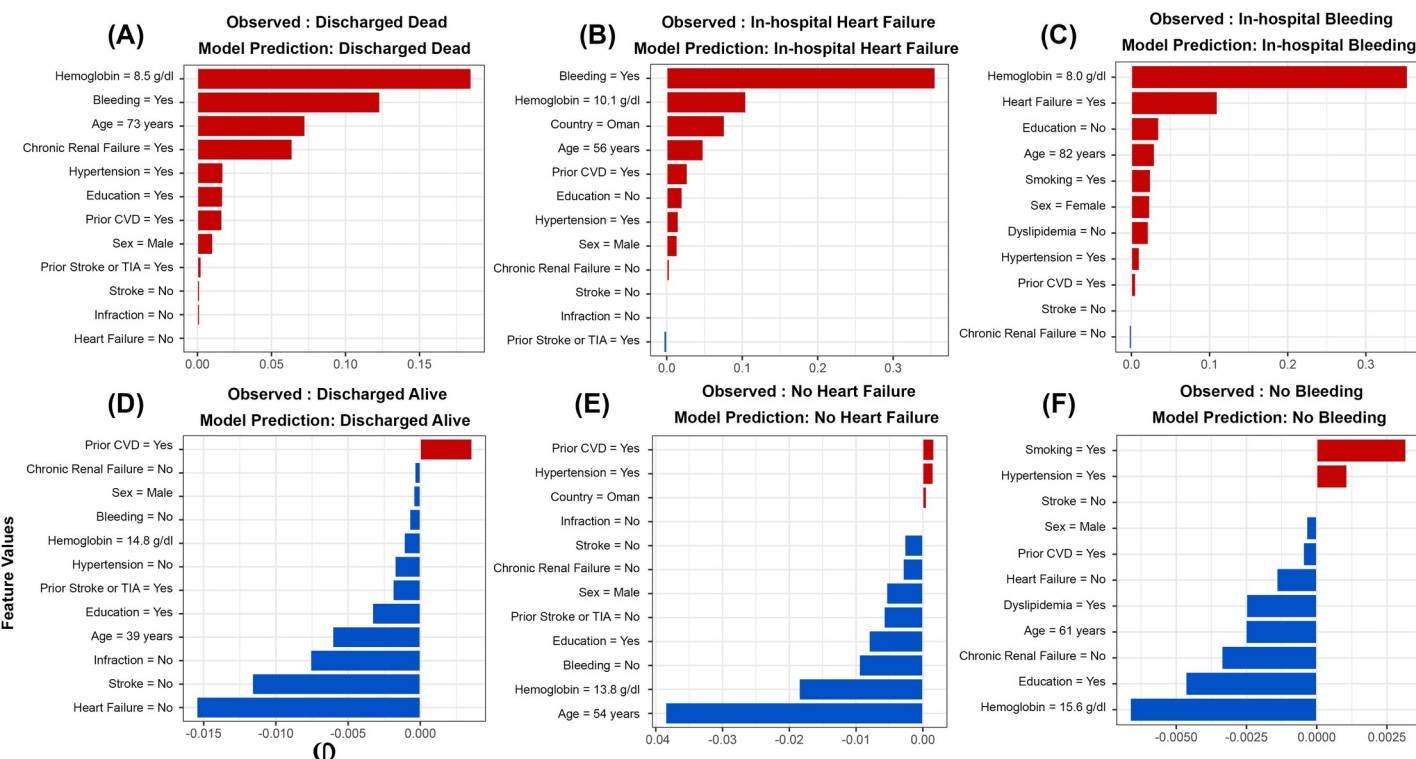

**Fig 4. Value contributions for the respective risk of acute coronary syndromes (ACS) based on Shapley Values (φ) for Six Individual Patients.** (A) a patient discharged dead; (B) a patient with an in-hospital heart failure; (C) a patient with an in-hospital bleeding; (D) a patient discharged alive; (E) a patient discharged alive with no in-hospital heart failure; and (F) a patient discharged alive with no in-hospital bleeding. Red bars indicate positive outcomes, and blue bars indicate negative outcomes. Positive Shapely value indicates that this feature increased the risk of ACS outcome, whereas negative values indicate that this feature decreased the risk of ACS outcome. The values next to each feature name indicate the observed value of that feature for that patient. *CVD: Cardiovascular disease; *TIA: Transient ischemic attack.

unique and complex role of anemia on admission in the prognosis of different ACS outcomes. Overall, we found that the initial hemoglobin values at admission were the most important variables shaping the risk of ACS in-hospital outcomes. Notably, the relationship between admission anemia and other baseline characteristics was non-linear in shaping the risk of in-hospital events. These rigorous interpretable insights generated by our ML approach can not only improve clinicians' prognostic efforts but assist with reducing the public health and economic implications of this important cardiovascular disease.

Our ML algorithms consistently identified admission hemoglobin values and in-hospital cardiovascular disease events as the most important predictors for the risk of in-hospital mortalities, heart failure, and bleeding (Fig 1). Our results were consistent with past studies in terms of the critical role of anemia in shaping the risk of ACS in-hospital outcomes [41,42]. We found that hemoglobin values on admission equal to or less than 10 g/dl increases the risk of in-hospital death, heart failure, and bleeding (Fig 2B–2D). This finding has also been observed in previous studies since anemia causes hypo-oxygenation to major organs, including the heart, leading to compromise of cardiac function and sterile inflammation, which accelerates atherosclerosis and promotes thrombosis [43]. Further, cICE plots (Fig 2D–2F) and feature interaction plots (Fig 3) show that the relationship between anemia and the risk of several ACS outcomes is non-linear and far more complex [20,42]. These results indicate that anemia on admission has both a direct and indirect role in the prognosis of ACS and that the combination of anemia and other baseline characteristics shaped the risk of in-hospital outcomes. While we were unable to quantify a distinct relationship between the risk of mortality and bleeding with age (Fig 2F), the individual interaction between mortality and age showed that patients aged greater than 75 years old with initial hemoglobin value less than 10 g/dl are more likely to die from ACS related complication (*i.e.*, darker shades of red tend to accumulate across the spectrum of hemoglobin values; Fig 3C). This also confirms the complex non-linear relationship between hemoglobin and age in shaping the risk of ACS events [44].

Furthermore, our models demonstrate that males are more likely to experience heart failure than females if initial hemoglobin values were less than 10 g/dl (Fig 3F), which agrees with the notion that sex is an important modifier and contributor to the development of heart failure [44]. We also show that anemic patients with chronic renal failure are more likely to develop a bleeding event (Fig 3I). This could be because a patient with chronic renal failure suffers from severe albuminuria, which is an important facilitator of bleeding events [45]. These findings are expected since anemia is often associated with other comorbidities such as infections, chronic inflammatory conditions, chronic renal failure and neoplastic diseases [42]. Yet, unlike past studies, we found a slight non-linear increase in the risk of ACS outcomes at hemoglobin thresholds greater than 15 g/dl, particularly in the mortality and bleeding models (Fig 2C and 2D). This U-shaped trend is notably distinct in the individual interactions between hemoglobin on one side and age, sex, and chronic renal failure on the other (Fig 3C, 3F and 3I). These findings strongly quantify the notion of the role of polycythemia in shaping the risk in-hospital ACS events with other risk factors in the higher dimensional space [46,47]. Indeed, past studies confirmed that polycythemia could cause both thrombosis and bleeding events in the same patients with ACS [48,49]. Additionally, polycythemia can cause ischemia which leads to the development of arterial or venous thrombosis, myocardial, or heart failure [49].

One important limitation of the Gulf COAST registry is the population size, and therefore generalizability of our findings might be biased toward the population included in our analyses. Yet, many of our ML inferences agrees with the findings of the GULF RACE II study in terms of the role of anemia in ACS patients, which included these three Gulf countries [23]. Nevertheless, ML predictive models are mainly meant to reveal complex relationships in the available data that might guide and improve future prognostic efforts of ACS events in the

same population where the data have been collected. While our predictive models have not been validated on recent ACS registries, the use of the k-fold cross-validation approach reduces the chances of overfitting and strengthens the validity of their subsequent inference. The inability to fit valid predictive models for other ACS related outcomes such as infarction/reinfarction, PCI, and stroke is another limitation of the present study since the number of the cases were substantially less than the other selected outcomes (Table 1). Indeed, the prevalence of PCI in the COAST registry was substantially low (0.35%; Table 1); this is due to the limited availability of facilities for such procedures in the selected countries, as well as most of the selected patients did not need that procedure [26]. Yet, future studies should focus on evaluating the impact of hemoglobin level on other common ACS events when sufficient data is available. It is worth noting that the number of mortalities (n = 167; Table 1) was substantially less than the heart failure events (n = 521; Table 1).

In contrast, the predictive power of our mortality model was remarkably higher than the heart failure model (Table 2). Therefore, our ML analytical pipeline is insensitive to the number of event outcomes in the dataset but can be more sensitive to selected features or to the way how the features were coded and calibrated [50]. Thus, our selected features were better predictors of motilities than other ACS events, and future efforts should attempt to either add other relevant features or calibrate the selected features to improve the performance of these predictive models.

The complexity of ACS epidemiology coupled with the increasing size of registry data, as well as the highly non-linear relationships between admission anemia, other baseline characteristics, and ACS in-hospital events, highlight the strength of our ML analytical pipeline. Our selected ML algorithms were shown to outperform commonly used algorithms such as logistic regression, as well as risk stratification tools like TIMI, EMMACE, and the GRACE models, due to their flexibility in quantifying non-linear relationships with minimal underlying statistical assumptions. [5,6,51]. Past ACS studies that used similar ML algorithms were mainly focused on comparing their predictive power (*i.e.*, black-box approach) to traditional risk stratification tools rather than their interpretability in a clinical setting [6,51]. Hence, providing an interpretable predictive model will further help to improve the in-hospital decision making and, ultimately, the overall prognosis of ACS. Thus, our study represents the first attempt to implement an interpretable ML pipeline focused on unveiling complex relationships in the higher dimensional space to improve clinicians' ACS prognostic efforts, particularly in the Middle East.

Further, we illustrated the remarkable applicability of Shapley values to elucidate in finer scales what each model means in terms of predicted risk of different ACS events (*e.g.*, why a specific patient developed an ACS outcome, while the other did not?). This unique and intuitive attribute can be used to improve in-hospital clinician's prognoses and subsequently reduce the implications of different ACS outcomes. For example, for a randomly selected patient who had been discharged dead (Fig 4A), having low initial hemoglobin values (< 10 g/dl) and bleeding put that patient at high risk of in-hospital death (probability > 0.8). Conversely, the other selected patient who was discharged alive (Fig 4D) completely lacks such risk factors. Thus, patients with a similar Shapley profile of the dead patient (Fig 4A) should be targeted with rigorous interventions to reduce the risk of in-hospital mortalities or other ACS events (Fig 4B and 4C). Finally, future studies of ML applications in clinical settings should explore such methods for resources allocation within health care systems [52]. For example, length of stay is an important outcome that requires substantial resources when the duration of patient stay in the hospital is long. Therefore, the ML model's predictive ability can help guide the mobilization of clinical resources to targeted patients that are expected to stay longer due to their clinical profile.

## Conclusion

This study represents a unique attempt to implement an interpretable ML pipeline focused on revealing the complex relationship between ACS events and the role of anemia in predicting multiple ACS outcomes. We showed that anemia was the most important predictor of mortality, heart failure, and bleeding and had remarkably non-linear relationships with both ACS outcomes and patients' baseline characteristics. We demonstrated how our ML pipeline outperformed commonly used statistical and risk stratification methods due to its minimal statistical assumptions and ability to elucidate the predicted risk of each individual patient based on their unique risk factors in finer scales. To the authors' knowledge, a fully interpretable ML pipeline has not been yet implemented widely in clinical settings, particularly in the Middle East. Therefore, our ML models can improve clinicians' prognostic efforts and be used to guide policymakers in reducing the burdens of ACS on public health and the economy worldwide.

## Author Contributions

**Conceptualization:** Faisal Alsayegh, Moh A. Alkhamis.

**Data curation:** Faisal Alsayegh, Fatima Ali, Sreeja Attur, Mohammad Zubaid.

**Formal analysis:** Moh A. Alkhamis, Sreeja Attur, Nicholas M. Fountain-Jones.

**Investigation:** Moh A. Alkhamis, Fatima Ali, Nicholas M. Fountain-Jones.

**Methodology:** Moh A. Alkhamis, Nicholas M. Fountain-Jones.

**Project administration:** Faisal Alsayegh, Fatima Ali, Mohammad Zubaid.

**Resources:** Faisal Alsayegh, Sreeja Attur, Mohammad Zubaid.

**Software:** Moh A. Alkhamis, Sreeja Attur.

**Supervision:** Faisal Alsayegh, Mohammad Zubaid.

**Validation:** Faisal Alsayegh, Moh A. Alkhamis.

**Writing – original draft:** Faisal Alsayegh, Moh A. Alkhamis, Nicholas M. Fountain-Jones, Mohammad Zubaid.

**Writing – review & editing:** Moh A. Alkhamis, Nicholas M. Fountain-Jones.

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
