## [Decision Letter · Decision Letter 0]

23 Jun 2021

PONE-D-21-13062

Machine Learning Reveals Deeper Insights into the Role of Anemia in the Outcome of Acute Coronary Syndrome

PLOS ONE

Dear Dr. Ali,

Thank you for submitting your manuscript to PLOS ONE. After careful consideration, we feel that it has merit but does not fully meet PLOS ONE’s publication criteria as it currently stands. Therefore, we invite you to submit a revised version of the manuscript that addresses the points raised during the review process.

We look forward to receiving your revised manuscript.

Kind regards,

Andreas Zirlik, MD

Academic Editor

PLOS ONE

Journal Requirements:

2. Please include the data sources used in the Data availability statement and Methods section. Please also indicate in the Data availability statement whether you are able to openly share the code used, and if so, where others can access this.

3. Please ensure that you include a title page within your main document. We do appreciate that you have a title page document uploaded as a separate file, however, as per our author guidelines (http://journals.plos.org/plosone/s/submission-guidelines#loc-title-page) we do require this to be part of the manuscript file itself and not uploaded separately.

4. Thank you for stating the following in the Financial Disclosure section:

"Gulf COAST is an investigator-initiated study, financially supported by AstraZeneca and sponsored and overseen by Kuwait University (Project Code XX02/11). Neither Kuwait University nor AstraZeneca had any role in study design, data collection, data analysis or writing of the manuscript."

We note that you received funding from a commercial source: AstraZeneca.

5. Please amend your list of authors on the manuscript to ensure that each author is linked to an affiliation. Authors’ affiliations should reflect the institution where the work was done (if authors moved subsequently, you can also list the new affiliation stating “current affiliation:….” as necessary).

Reviewers' comments:

Reviewer's Responses to Questions

**Comments to the Author**

1. Is the manuscript technically sound, and do the data support the conclusions?

Reviewer #1: No

Reviewer #2: Partly

2. Has the statistical analysis been performed appropriately and rigorously? 

Reviewer #1: No

Reviewer #2: I Don't Know

3. Have the authors made all data underlying the findings in their manuscript fully available?

Reviewer #1: No

Reviewer #2: No

4. Is the manuscript presented in an intelligible fashion and written in standard English?

Reviewer #1: No

Reviewer #2: Yes

5. Review Comments to the Author

Reviewer #1: The Authors present the following aims of their study:

1) To identify factors associated with in-hospital adverse events post ACS

2) To explore the role of haemoglobin values at admission and their interaction with other relevant factors

3) To explore the role of patients’ initial hemoglobin values at admission, and their interactions with other relevant factors for each selected ACS adverse event.

4) To extend and evaluate the created models in the context of in-hospital individual level prognosis.

The authors found in their analysis that the strongest predictor for the in-hospital adverse events death, heart failure and bleeding was length of hospital stay. Haemoglobin was the second strongest predictor with strong interaction with age for mortality and sex for bleeding and heart failure. Shapley values for the contribution of each predictor in 6 selected patients confirm the explanatory strength of length of stay and haemoglobin.

The individual contribution of various predictors to explain short term prognosis after ACS is definitely an important topic to study. It might improve patient management and provide valuable insights into the pathophysiology of adverse events.

Furthermore, machine learning approaches are quite new to cardiovascular risk prediction and need to find their optimal place in the field.

However, the authors are advised to completely rethink the presented analysis.

By their machine learning approach the authors found that length of hospitalization and haemoglobin at admission were the most important features for predicting the risk of mortality, heart failure, and bleeding. They further report that “risk of all ACS outcomes increases when patients stay in the hospital for more than 10 days“.

This is one of the major limitations of the study: Length of hospital stay is an outcome parameter rather than a predictor and should not be used as a predicting parameter for events during the same hospitalization (the authors count their outcome parameters death, heart failure, bleeding only as long as the patient is still in hospital). In case of death, this event ends hospitalization and thus has a direct influence on its predictive parameter length of stay. Similarly, patients experience their in-hospital heart failure and in-hospital bleeding before the value of their predicting variable “length of hospital stay” can be known. In addition, in-hospital heart failure and bleeding have a direct influence on length of stay: In most patients they would prolong hospitalization so that length of stay is a function of heart failure and bleeding and not vice versa.

This is in my opinion misconception that also invalidates further calculations which are based on length of stay being among the three strongest predictors.

Therefore, I do believe that this manuscript cannot be accepted for publication in the Journal.

Further comments:

The authors need to elaborate more on the rationale of the research question and the use of machine learning for their specific research question.

The authors should put more emphasis in explaining the clinical impact of their results and the use of machine learning tools.

The authors should revise the manuscript – including language - to improve readability. The manuscript text is interrupted by a table an various figure legends, which hampers understanding such a complex analysis. Furthermore, I did not have access to supplemetary table 1.

Figure 2: The authors state that the risk for death, heart failure, and bleeding markedly increases after 10 days of hospital stay. I can confirm this statement for heart failure and bleeding. However, for mortality it looks like a steady rather than an abrupt increase.

The authors state that the risk of heart failure increases at the age of 50 years and above. However, in figure 2H it looks as if the risk of heart failure does not increase below 60 years and markedly increase around 70 years and older.

The Authors state that patients with Hb levels <10g/dl are at a higher risk of death, heart failure, or bleeding. The risk of heart failure, however, increases already at higher haemoglobin levels ( <13g/dl?). See also Figures 3 F and 3 I. The authors should comment on these discrepant findings.

I believe figure 2 would benefit from using the same scale on all y-axes for better comparison.

Reviewer #2: In the submitted study entitled „Machine Learning Reveals Deeper Insights into the Role of Anemia in the Outcome of Acute Coronary Syndrome” the authors used a multi-algorithm machine learning ensemble pipeline on data of 4044 patients from the Gulf-COAST registry which included patients with the diagnosis of ACS between January 2012 and January 2013.

If outcomes or complications of a patient presenting with ACS could be anticipated on patient characteristics from admission or the clinical course, a more patient tailored approach could lead to better outcomes. Therefore the approach to use a machine learning algorithm on the existing registry seems intriguing.

However there are some major concerns regarding the submitted study.

One major concern is that the study seems to lack a specific hypothesis and scientific question and therefor is more a report of multiple incoherent findings rather than a comprehensive paper. In the abstract and the conclusion the authors heavily emphasize their findings on anemia and ACS and claim that they “revealed the unique and complex role of anemia on admission in the prognosis of different ACS outcomes”. However the majority of the presented data are on the role of length of hospital stay and age. Furthermore the findings on anemia are not reported or discussed in sufficient depth. In regard to Figure 2 the authors state “[…] patients with initial hemoglobin values of equal or less than 10 g/dl are more likely to be discharge dead, had heart failure or develop bleeding”. While this seems to be true for mortality (Fig 2 D), the curve in Figure 2 E for heart failure seems to be u-shaped with a high risk at an Hb around 10 but lower risk at Hb 5-9 and 11-15, which is not addressed in the manuscript.

Another major concern is with the underlying data which were analyzed. In table 1 the number of Patients receiving PCI is reported which is 0.35 % of the 4044 Patients with ACS. This rate of PCIs in an ACS collective seems surprisingly low. Also the type of ACS is not further evaluated in the manuscript. I think it would enhance the quality of the manuscript to evaluate the impact of the different hemoglobin levels in the different types of ACS. In addition it would be important to get information on how many values actually were present in regard to the single characteristics and outcomes.

In addition there are some minor questions:

- Line 142: “Heart failure was the third most important feature for predicting ACS mortalities […]”

Does heart failure in this context mean heart failure as a preexisting condition or a complication?

- Line 205-207

The authors discuss that length of stay is probably dependent on complications instead of being an independent factor toward predicting the risks of ACS. I agree with the authors but I think that raises the question why length of stay is addressed as a risk factor and not as an outcome. I think it would make more sense to analyze it in that way.

- Figure I

In Figure I the Abbreviation CRF is used without being explained in the manuscript

- Figure 3 B, E, H

Please explain more detailed what those graphs are actually showing.

Overall I think the manuscript should be revised de novo with a more comprehensible story.

6. PLOS authors have the option to publish the peer review history of their article (what does this mean?). If published, this will include your full peer review and any attached files.

Reviewer #1: No

Reviewer #2: No

---

## [Author Response · Author response to Decision Letter 0]

1 Oct 2021

Reviewer #1: 

1- However, the authors are advised to completely rethink the presented analysis.

By their machine learning approach the authors found that length of hospitalization and haemoglobin at admission were the most important features for predicting the risk of mortality, heart failure, and bleeding. They further report that “risk of all ACS outcomes increases when patients stay in the hospital for more than 10 days“. This is one of the major limitations of the study: Length of hospital stay is an outcome parameter rather than a predictor and should not be used as a predicting parameter for events during the same hospitalization (the authors count their outcome parameters death, heart failure, bleeding only as long as the patient is still in hospital). In case of death, this event ends hospitalization and thus has a direct influence on its predictive parameter length of stay. Similarly, patients experience their in-hospital heart failure and in-hospital bleeding before the value of their predicting variable “length of hospital stay” can be known. In addition, in-hospital heart failure and bleeding have a direct influence on length of stay: In most patients they would prolong hospitalization so that length of stay is a function of heart failure and bleeding and not vice versa. This is in my opinion misconception that also invalidates further calculations which are based on length of stay being among the three strongest predictors. Therefore, I do believe that this manuscript cannot be accepted for publication in the Journal.

- We agree with the reviewer’s point in terms of excluding length of stay from the analysis. Hence, we have repeated the whole analysis and revised all the results and figures. While removing length of stay slightly dropped the performance of all selected machine learning algorithms, the initial hemoglobin levels remained as the strongest predictor and interacting variable with other predictors. Therefore, we hope that the reviewer now sees the revised version acceptable for publication and thank them for their valuable time in reviewing the manuscript.

2- The authors need to elaborate more on the rationale of the research question and the use of machine learning for their specific research question.

- We elaborated extensively in both introduction (lines 45-74, 101-106 & 107-116) and discussion sections (lines 326-329, & 347-359) for the rational of using ML for our specific research question as suggested by the reviewer.

3- The authors should put more emphasis in explaining the clinical impact of their results and the use of machine learning tools.

- We extensively revisited the explanation of the clinical impact of our results derived from ML Model as suggested by the reviewer. 

4- The authors should revise the manuscript – including language - to improve readability. The manuscript text is interrupted by a table an various figure legends, which hampers understanding such a complex analysis. Furthermore, I did not have access to supplemetary table 1.

- We extensively revised the linguistics of the manuscript as suggested by the reviewer. However, the presence of tables and figure legends in the body of the manuscript is due to following the journal’s strict submission guidelines. We also apologize for miss-referencing supplementary table 1 as it was referring to table 1.

5- Figure 2: The authors state that the risk for death, heart failure, and bleeding markedly increases after 10 days of hospital stay. I can confirm this statement for heart failure and bleeding. However, for mortality it looks like a steady rather than an abrupt increase.

- The point is no longer valid, after removing length of stay from the analysis as suggested by the reviewer.

6- The authors state that the risk of heart failure increases at the age of 50 years and above. However, in figure 2H it looks as if the risk of heart failure does not increase below 60 years and markedly increase around 70 years and older.

- After removing length of stay from the analysis the results slightly changed, and we commented extensively on this point as suggested by the reviewer (lines 299-305).

7- The Authors state that patients with Hb levels <10g/dl are at a higher risk of death, heart failure, or bleeding. The risk of heart failure, however, increases already at higher haemoglobin levels ( <13g/dl?). See also Figures 3 F and 3 I. The authors should comment on these discrepant findings.

- We extensively commented on this as suggested by the reviewer (Lines 313-322).

8- I believe figure 2 would benefit from using the same scale on all y-axes for better comparison.

- We attempted to unify the y-axes for all figures as suggested by the reviewer. However, some of the resulting figures looked distorted with large empty spaces, as the probability thresholds differs between important features. Yet, the x-axis is unified among the same features.

 

Reviewer #2 (note that revisions starts after the new figures below): 

1- One major concern is that the study seems to lack a specific hypothesis and scientific question and therefor is more a report of multiple incoherent findings rather than a comprehensive paper. In the abstract and the conclusion the authors heavily emphasize their findings on anemia and ACS and claim that they “revealed the unique and complex role of anemia on admission in the prognosis of different ACS outcomes”. However the majority of the presented data are on the role of length of hospital stay and age. 

- We made substantial revisions throughout the manuscript in this regard. We made the paper more oriented toward how machine learning is more effective than classical risk stratification tools used by most clinicians in revealing complex relationships between different risk factors and ACS in-hospital events. Then how our selected ML models can produce rich outputs that is more interpretable than past used ML models. Finally, we showed how our ML untangled the complex relationship between initial hemoglobin values at admission and different common in-hospital ACS events with an extensive and elaborated discussion. We hope that the current version is more acceptable to the reviewer and we thank them for their time in reviewing the manuscript and provide useful suggestions. 

2- Furthermore the findings on anemia are not reported or discussed in sufficient depth. In regard to Figure 2 the authors state “[…] patients with initial hemoglobin values of equal or less than 10 g/dl are more likely to be discharge dead, had heart failure or develop bleeding”. While this seems to be true for mortality (Fig 2 D), the curve in Figure 2 E for heart failure seems to be u-shaped with a high risk at an Hb around 10 but lower risk at Hb 5-9 and 11-15, which is not addressed in the manuscript.

- We extensively commented on this as suggested by the reviewer (lines 313-322).

3- Another major concern is with the underlying data which were analyzed. In table 1 the number of Patients receiving PCI is reported which is 0.35 % of the 4044 Patients with ACS. This rate of PCIs in an ACS collective seems surprisingly low. Also the type of ACS is not further evaluated in the manuscript. I think it would enhance the quality of the manuscript to evaluate the impact of the different hemoglobin levels in the different types of ACS. In addition it would be important to get information on how many values actually were present in regard to the single characteristics and outcomes.

- We did comment on this in the discussion section as suggested by the reviewer (lines 331-340). Further, the evaluation of other types of ACS will make the content of the manuscript complicated and distracting to the reader. Yet, we commented on this in the discussion section. Thus, we think that current version of the manuscript is already rich and extensive.

4- Line 142: “Heart failure was the third most important feature for predicting ACS mortalities […]” Does heart failure in this context mean heart failure as a preexisting condition or a complication?

- It is an in-hospital heart failure which different from prior CVD. We clarified this and made it distinct throughout the manuscript.

5- Line 205-207: The authors discuss that length of stay is probably dependent on complications instead of being an independent factor toward predicting the risks of ACS. I agree with the authors but I think that raises the question why length of stay is addressed as a risk factor and not as an outcome. I think it would make more sense to analyze it in that way.

- We removed length of stay from the models as suggested by reviewer 1 and repeated the whole analysis. Yet, we agree with the reviewer’s notion of setting length of stay as an outcome. However, the manuscript is already rich in content after the substantial revision. Thus, building a model for length of stay might complicate the results and distract the reviewer. Yet, we commented on this in the discussion section (Lines 367-372). 

6- Figure I In Figure I the Abbreviation CRF is used without being explained in the manuscript

- Abbreviation CRF has been spelled out in all of the figures as suggested by the reviewer.

7- Figure 3 B, E, H; Please explain more detailed what those graphs are actually showing.

- We described and interpreted these figures as suggested by the reviewer.

---

## [Decision Letter · Decision Letter 1]

17 Nov 2021

PONE-D-21-13062R1Anemia or other comorbidities? Using machine learning to reveal deeper insights into the drivers of in-hospital Acute coronary syndromesPLOS ONE

Dear Dr. Alkhamis,

Thank you for submitting your manuscript to PLOS ONE. After careful consideration, we feel that it has merit but does not fully meet PLOS ONE’s publication criteria as it currently stands. Therefore, we invite you to submit a revised version of the manuscript that addresses the points raised during the review process. Most of the points are formal points that need to be addressed.

We look forward to receiving your revised manuscript.

Kind regards,

Andreas Zirlik, MD

Academic Editor

PLOS ONE

Journal Requirements:

Reviewers' comments:

Reviewer's Responses to Questions

**Comments to the Author**

1. If the authors have adequately addressed your comments raised in a previous round of review and you feel that this manuscript is now acceptable for publication, you may indicate that here to bypass the “Comments to the Author” section, enter your conflict of interest statement in the “Confidential to Editor” section, and submit your "Accept" recommendation.

Reviewer #2: All comments have been addressed

2. Is the manuscript technically sound, and do the data support the conclusions?

Reviewer #2: Yes

3. Has the statistical analysis been performed appropriately and rigorously? 

Reviewer #2: I Don't Know

4. Have the authors made all data underlying the findings in their manuscript fully available?

Reviewer #2: Yes

5. Is the manuscript presented in an intelligible fashion and written in standard English?

Reviewer #2: Yes

6. Review Comments to the Author

Reviewer #2: In the revised manuscript the authors made substantial changes to both the analysis (by excluding length of stay and rerunning the analysis) and the writing of the text. Overall the applied changes do significantly enhance the quality of the manuscript. There are only some minor comments which I would like to make:

1) The new title of the manuscript is conflicting. “Using machine learning to reveal deeper insights into the drivers of in-hospital Acute coronary syndromes” indicates that the examined population did suffer from ACS during an hospitalization rather than being admitted for ACS. Please reframe to avoid confusion.

2) Line 72, 93 and 349: You refer to a risk stratification tool called MACE. To my knowledge MACE is a composite of major adverse cardiac events and not a risk stratification tool. In the reference that you added there is no risk stratification tool called “MACE” descripted. Please clarify what you are referring to.

3) In the abstract you claim that “anemia was the most important predictor of mortality,

32 heart failure, and bleeding […]” however in your results you say that “in-hospital heart failure followed by initial hemoglobin values at admission and age, were the most important features for predicting the risk of mortality”. Please correct that in the abstract.

4) You explained the low rates of PCI in your ACS-registry (0,35%) with accessibility and lack of necessity in most patients. Please report the rates of the subtypes of ACS in your baseline characteristics as it is very important to understand the quality of care the examined patients received, in regard to generalizability of your findings.

5) In table 1 the reference-letter for the explanation of CVD in the figure legend is missing. Typically CVD also is the abbreviation for cardiovascular disease. If you really mean coronary vascular disease = coronary artery disease the correct abbreviation would be CAD.

7. PLOS authors have the option to publish the peer review history of their article (what does this mean?). If published, this will include your full peer review and any attached files.

Reviewer #2: No

---

## [Author Response · Author response to Decision Letter 1]

23 Nov 2021

Reviewer #2: 

In the revised manuscript, the authors made substantial changes to both the analysis (by excluding length of stay and rerunning the analysis) and the writing of the text. Overall the applied changes do significantly enhance the quality of the manuscript. There are only some minor comments which I would like to make:

- We thank the reviewer for their valuable time and comments, which substantially improved the quality of the manuscript and hope that the minor requested revisions were fulfilled as suggested. 

1) The new title of the manuscript is conflicting. “Using machine learning to reveal deeper insights into the drivers of in-hospital Acute coronary syndromes” indicates that the examined population did suffer from ACS during an hospitalization rather than being admitted for ACS. Please reframe to avoid confusion.

- We fixed the title as suggested by the reviewer, and we welcome additional suggestions to improve the title of the manuscript. 

2) Line 72, 93 and 349: You refer to a risk stratification tool called MACE. To my knowledge MACE is a composite of major adverse cardiac events and not a risk stratification tool. In the reference that you added there is no risk stratification tool called “MACE” described. Please clarify what you are referring to.

- We agree with the reviewer’s comment, therefore to avoid this confusion, we replaced it with other common risk stratification tools such EMMACE and GRACE with their relevant citations as examples. 

3) In the abstract, you claim that “anemia was the most important predictor of mortality,32 heart failure, and bleeding […]” however, in your results, you say that “in-hospital heart failure followed by initial hemoglobin values at admission and age, were the most important features for predicting the risk of mortality”. Please correct that in the abstract.

- We fixed the sentence in the abstract as suggested by the reviewer.

4) You explained the low rates of PCI in your ACS registry (0,35%) with accessibility and lack of necessity in most patients. Please report the rates of the subtypes of ACS in your baseline characteristics as it is very important to understand the quality of care the examined patients received in regard to the generalizability of your findings.

¬- It is very important to emphasize that the reported PCI in table 1 is an outcome PCI. In other words, this was the PCI rate for patients who during hospitalization suffered recurrent infarction as a complication during their hospital stay. Therefore, the rate of 0.35% is for the 14 patients whose denominator was 71 patients (who suffered re-infarction during their hospital stay). In other words, 14 of 71 (19.7%). That said, we added the ACS subtypes in table 1 as suggested by the reviewer.

5) In table 1, the reference letter for the explanation of CVD in the figure legend is missing. Typically CVD also is the abbreviation for cardiovascular disease. If you really mean coronary vascular disease = coronary artery disease, the correct abbreviation would be CAD.

¬- We agree with the reviewer’s comment, here, we were using the expression CVD as an abbreviation for cardiovascular disease and therefore, we fixed it in the captions of table 1, as well as of figures 1, 3 and 4.

---

## [Editor Report · Decision Letter 2]

11 Jan 2022

Anemia or other comorbidities? Using machine learning to reveal deeper insights into the drivers of acute coronary syndromes in hospital admitted patients

PONE-D-21-13062R2

Dear Dr. Alkhamis,

We’re pleased to inform you that your manuscript has been judged scientifically suitable for publication and will be formally accepted for publication once it meets all outstanding technical requirements.

Kind regards,

Andreas Zirlik, MD

Academic Editor

PLOS ONE
---

## [Editor Report · Acceptance letter]

13 Jan 2022

PONE-D-21-13062R2 

Anemia or other comorbidities? Using machine learning to reveal deeper insights into the drivers of acute coronary syndromes in hospital admitted patients 

Dear Dr. Alkhamis:

I'm pleased to inform you that your manuscript has been deemed suitable for publication in PLOS ONE. Congratulations! Your manuscript is now with our production department. 

Kind regards, 

on behalf of

Univ. Prof. Dr. Andreas Zirlik 

Academic Editor

PLOS ONE